# Clinical-Grade Patches as a Medium for Enrichment of Sweat-Extracellular Vesicles and Facilitating Their Metabolic Analysis

**DOI:** 10.3390/ijms24087507

**Published:** 2023-04-19

**Authors:** Syeda Tayyiba Rahat, Mira Mäkelä, Maryam Nasserinejad, Tiina M. Ikäheimo, Henna Hyrkäs-Palmu, Rasmus I. P. Valtonen, Juha Röning, Sylvain Sebert, Anni I. Nieminen, Nsrein Ali, Seppo Vainio

**Affiliations:** 1Laboratory of Developmental Biology, Faculty of Biochemistry and Molecular Medicine, University of Oulu, 90220 Oulu, Finland; 2Research Unit of Population Health Research, Faculty of Medicine, University of Oulu, 90570 Oulu, Finland; 3Infotech Oulu, University of Oulu, 90014 Oulu, Finland; 4Department of Community Medicine, University of Tromsø, N-9037 Tromsø, Norway; 5Research Unit of Population Health, University of Oulu, 90220 Oulu, Finland; 6Research Unit of Biomedicine, Medical Research Center, Faculty of Medicine, University of Oulu, Oulu University Hospital, 90220 Oulu, Finland; 7Biomimetics and Intelligent Systems Group, Faculty of Information Technology and Electrical Engineering, University of Oulu, 90570 Oulu, Finland; 8FIMM Metabolomics Unit, Institute for Molecular Medicine Finland, University of Helsinki, 00014 Helsinki, Finland; 9Flagship GeneCellNano, University of Oulu, 90220 Oulu, Finland; 10Kvantum Institute, University of Oulu, 90014 Oulu, Finland

**Keywords:** sweat, EVs, clinical grade patch, metabolomic, BMI, blood glucose

## Abstract

Cell-secreted extracellular vesicles (EVs), carrying components such as RNA, DNA, proteins, and metabolites, serve as candidates for developing non-invasive solutions for monitoring health and disease, owing to their capacity to cross various biological barriers and to become integrated into human sweat. However, the evidence for sweat-associated EVs providing clinically relevant information to use in disease diagnostics has not been reported. Developing cost-effective, easy, and reliable methodologies to investigate EVs’ molecular load and composition in the sweat may help to validate their relevance in clinical diagnosis. We used clinical-grade dressing patches, with the aim being to accumulate, purify and characterize sweat EVs from healthy participants exposed to transient heat. The skin patch-based protocol described in this paper enables the enrichment of sweat EVs that express EV markers, such as CD63. A targeted metabolomics study of the sweat EVs identified 24 components. These are associated with amino acids, glutamate, glutathione, fatty acids, TCA, and glycolysis pathways. Furthermore, as a proof-of-concept, when comparing the metabolites’ levels in sweat EVs isolated from healthy individuals with those of participants with Type 2 diabetes following heat exposure, our findings revealed that the metabolic patterns of sweat EVs may be linked with metabolic changes. Moreover, the concentration of these metabolites may reflect correlations with blood glucose and BMI. Together our data revealed that sweat EVs can be purified using routinely used clinical patches, setting the foundations for larger-scale clinical cohort work. Furthermore, the metabolites identified in sweat EVs also offer a realistic means to identify relevant disease biomarkers. This study thus provides a proof-of-concept towards a novel methodology that will focus on the use of the sweat EVs and their metabolites as a non-invasive approach, in order to monitor wellbeing and changes in diseases.

## 1. Introduction

Sweat is a bodily fluid that is secreted continuously and imperceptibly by the body; analogous to the use of urine, saliva, and tears in clinical diagnoses, sweat could offer a medium for measuring, non-invasively, homeostasis in both healthy and disease conditions. Sweat is excreted by the sweat glands, three types of which have been characterized, and are classified as eccrine, apocrine, and apoeccrine glands. These glands differ in their function, distribution, and also their cellular structure [1]. Sweat plays a key role in maintaining homeostasis via thermoregulation of the body during physical activity, or when exposed to heat. However, it provides a critical barrier and protects the body against microbes [1,2]. Recent reports have started to point towards an extended role of the molecular constituents of sweat, such as urea, certain ions or salt concentration, raising possibilities that the skin may also offer a novel means of monitoring organ functions based on their secreted analytes or the response of skin cells to them [3,4,5,6].

The extracellular vesicles (EVs) that are widely secreted by cells of most metabolically active species have recently been observed in sweat and thereby evoked attention. EVs are categorized according to their cellular origin and size. Exosomes are 10–150 nm in diameter and are assembled in the multivesicular bodies (MVBs). The macrovesicles are 500–1000 nm in diameter and are derived directly from the donor cell plasma membrane. Both EV types are released into the extracellular space and transported throughout the body via bodily fluids [7].

The diagnostic value of EVs is based on the diverse array of molecular components contained within them: RNA, DNA, proteins, lipids, glycans and metabolites. These distinct properties of EVs offer a foundation for characterizing the molecular signatures of common diseases [8,9,10,11]. Recent reports have proposed sweat EVs as a valuable non-invasive approach for developing novel diagnostic solutions [12,13]. Thus, sweat and its components, including those of the EVs, may offer a non-invasive route to access a pool of diagnostic analytes; however, methods for collecting sweat cost-effectively from patient cohorts, such as diabetes for biomarker discoveries, remain elusive.

We describe a novel methodology to collect, isolate and characterize sweat EVs from healthy and Type 2 diabetes (T2D) participants in response to transient heat exposure, using clinical grade skin patches. We also show that sweat-enriched EVs express the EV biomarker CD63. Targeted metabolomic analysis of the sweat patch-enriched EVs revealed 24 metabolites, as determined by chromatography analysis. Analyzing metabolite’ signatures from healthy and Type 2 diabetes groups, following heat exposure, indicates a moderate increase of a few metabolites in T2D sweat EVs when compared with sweat EVs obtained from healthy participants. These data suggest that metabolite levels from sweat EVs may serve as a means for reflecting metabolic changes in healthy and disease states.

Moreover, the excretion of sweat EVs and quantification of their constituent metabolites, when analyzed with cross-reference to blood glucose values or to BMI in healthy and Type 2 diabetes participants, highlights a strategy towards large-scale citizen science solutions and establishing an evidence-based medicine program for specific diseases, such as diabetes.

## 2. Results

### 2.1. Sweat Extracellular Vesicles (EVs) Can Be Enriched and Isolated via Clinical Grade Patches

To examine the presence of EVs in sweat using clinically available tools, we decided to assess if sweat-associated EVs could be collected with the aid of skin patches. For this purpose, we recruited 11 healthy volunteers to participate in heat exposure trials. The selection criteria of the subjects were: age, sex, blood glucose levels, and BMI (Table 1).

The participants were exposed to heat (+40 °C, relative humidity 20–40%) in a climatic chamber for 90 min, while resting. Prior to the trial, Sorbact^®^ patches (10 × 10 cm) were weighed and adhered to (15 × 15 cm) Tegaderm ^TM^ transparent film to cover the patch, thus preventing sweat loss and providing adhesion. The prepared patches were then attached to participants’ skin, on their back. Following 90 min of heat exposure, the patches were carefully removed, the adhesive film was separated and discarded, and the final weight of the patches was recorded. The absorbent layer of the patch was separated by cutting off the patch edges with scissors. Phosphate-buffered saline solution was added to the patch and incubated for 5 min. The mixture was first passed through A 40 µm sieve and then filtered through a 0.8 µm vacuum filtration unit. The filtrate was ultra-centrifuged at 100,000× *g* overnight at 4 °C to pull down the sweat EVs. The supernatant was discarded the following day, and the sweat EV’s pellet was washed using 1× PBS followed by centrifugation (100,000× *g* g^−2^ h) to remove any contamination, and was finally resuspended in 1× PBS. The procedure is described in Figure 1.

To characterize the properties of sweat EVs—such as their size, concentration (particles/mL), and the expression of various molecular components—we employed the Nano-particle Tracking Assay (NTA), electron microscopy (EM), ExoView, and Western blot methodologies. The NTA data revealed that sweat EVs were 100–600 nm in diameter, with EVs that were 150–250 nm in diameter being the most predominant (Figure 2A). NTA analysis also enabled an estimation of the overall quantity of enriched EVs in the patches from the different participants, with an average number being 6.69 × 10^10^ ± 4.58 × 10^10^ particles/mL (Table 2).

In line with the NTA data, subjecting sweat EVs to EM-based analysis confirmed that the EVs differed in size (Figure 2B). The ExoView-based studies, which enable the expression of a single protein to be detected and at the level of an individual EV, demonstrated that sweat EVs dominantly express the CD63 (red with 77%), the CD9 (green with 17%), and the CD81 (blue with 15%) EV biomarkers (Figure 2C,D, and Appendix A, respectively).

Immunoelectron microscopy using an antibody against the CD63 EV biomarker, which is widely expressed in EVs, confirmed the presence of CD63 (Figure 2E). Consistent with EM observations, Western blot analysis on two samples of sweat EVs isolated from different participants detected a strong band at 54 kDa, corresponding to CD63 (Lanes 3 and 4), while no band was detected in control samples (Lane 1: non-enriched sweat patch that was not attached to a participant, the contents of which were extracted and isolated following the same protocol of the sweat-enriched patches, and lane 2: 1× PBS) (Figure 2F and Appendix A).

In summary, our study describes a successful methodology for enriching and isolating EVs from sweat, collected non-invasively using clinically-approved patches.

### 2.2. Sweat Extracted EVs Express Cargo Metabolites

Sweat as a solute provides a means to measure sodium, lactate, potassium, and glucose components [14,15]. However, at present, no data are available to provide evidence of the presence of metabolites in sweat EVs, or whether their concentrations can be estimated. To investigate the metabolic composition of sweat EVs isolated from healthy participants following heat exposure, targeted global metabolomics profiling of 41 metabolites was performed.

Based on quality control (QC) of the samples (%CV < 20) and blank (% area < 20), 24 metabolites were identified by chromatography analysis, as illustrated in the heatmap (Figure 3A). Closer analysis of the values obtained in the heatmap indicated differences in the level of noted metabolites between healthy participants.

The metabolites enriched from sweat EVs were classified into subgroups. This categorization was based on their association with specific biochemical metabolic pathways, such as those producing amino acids, glutamate and fatty acids, as well as the TCA cycle and the glycolysis pathway. More specifically, metabolites from the amino acid pathway included alanine, aspartate, arginine, glycine, leucine, isoleucine, serine, proline, threonine, tyrosine, tryptophan, methionine, valine, and lysine. Those metabolites identified from glutathione and glutamate metabolism pathways included pyroglutamate, glutamate, and glutamine, while those arising from the fatty acid pathway included linoleate, myristate, and palmitate. Metabolites from the TCA cycle pathway included malate and succinate. Finally, a notable component from the glycolysis pathway was lactate, which is a well-established metabolite used for monitoring physical activity during sport (Figure 3A, and Appendix A).

We next aimed to compare, quantitively, the differences between the metabolites that were present in sweat EVs isolated from the eleven participants. The metabolite pattern of healthy participants’ sweat EVs was visualized with metabolite intensity heatmap, showing a clear abundance of central carbon metabolites (Figure 3A, and Appendix A).

Furthermore, the average metabolite abundance within the healthy participants group demonstrated large differences in metabolite production between the healthy participants (Figure 3B and Figure 4A–C).

Our data suggest that the metabolites of sweat EVs may provide a useful approach to assess the individualized response to heat exposure.

### 2.3. Metabolite Levels in EVs Extracted from Sweat May Provide a Means to Study their Association with Diseases

To reveal whether the metabolites from sweat EVs may represent a non-invasive solution for studying the metabolic changes between healthy and diseased groups, we recruited 10 volunteers diagnosed with Type 2 diabetes (T2D) to participate in the heat-exposure trial. The selection criteria of the participants accounted for age, sex, blood glucose levels, and BMI (Table 3).

To investigate the metabolic signatures of sweat EVs isolated from T2D participants following heat exposure, the 24 metabolites previously identified, and referred to in Figure 3 and Figure 4, were analyzed. As demonstrated in the heatmap, high variations of each metabolite’s levels in sweat EVs between different individuals within T2D group were observed (Figure 5A).

To compare the differences between each metabolite level between the T2D- and healthy-sweat EVs, the metabolite intensity (peak area) fold-changes were calculated after normalization to the total EVs concentration (particles/mL) and to the negative control. Of the metabolites analyzed, the levels of amino acids (arginine, asparagine) and fatty acids (linoleate, palmitate, and myristate) were elevated in T2D-sweat EVs compared to their control counterparts, from healthy individuals (Figure 5B–F).

Our data suggest, as a proof-of-concept, that the metabolites of sweat EVs may serve to reflect metabolic changes in a disease state, however, conducting large population size studies is required to further confirm this observation.

### 2.4. Association of the Metabolite Levels in Sweat EVs with Blood Glucose Level

To assess a possible correlation between blood glucose, and the abundance of metabolites in sweat EVs from healthy and T2D participants that had accumulated into the skin patches during heat exposure, we applied a Spearman’s rank association. Our data demonstrated that the concentration of EVs isolated from the sweat of healthy participants was significantly inversely associated with blood glucose (Table 4). However, a direct correlation, although not significant, between blood glucose and the concentration of EVs isolated from T2D-sweat group was observed (Table 5).

In the healthy group, the concentration of 18 metabolites (alanine, arginine, leucine, isoleucine, serine, proline, threonine, tryptophan, methionine, valine, lysine, pyroglutamate, glutamate, malate, and succinate) was significantly and indirectly correlated with blood glucose levels. However, the eight remaining metabolites—including glycine, tyrosine, aspartate, lactate, glutamine, linoleate, myristate, and palmitate—showed a trend that was inversely related to blood glucose (Table 4).

In contrast, in the T2D-sweat EVs group, the majority of the metabolites’ levels were directly correlated with blood glucose upon heat exposure, although not significantly. However, a few metabolites were inversely correlated with blood glucose, but not significantly, including glycine, lactate, proline, malate, myristate, palmitate, and aspartate (Table 5).

Overall, our results suggest that metabolites present in sweat EVs appear to correlate with blood glucose levels, however, given the limited number of participants, further validation is needed.

### 2.5. Changes in the Metabolites Contained in Sweat EVs in Relation to BMI

BMI is a standard health parameter that may indicate the risk of obesity, diabetes, or other metabolic disorders. Thus, to address the association between metabolite levels from sweat EVs and BMI, Spearman’s rank association was employed.

Our results revealed that there was no correlation between concentrations of EVs isolated from the sweat of healthy participants and BMI (Table 6). Moreover, when testing the association between BMI and metabolite levels in the EVs, no evidence of such association was found. Among the 24 metabolites identified in sweat EVs, 12 (pyroglutamate, glycine, alanine, arginine, asparagine, leucine, glutamate, methionine, serine, threonine, tyrosine, and tryptophan) showed a positive correlation with the BMI, but this was not statistically significant. However, eight of the metabolites showed a trend of negative correlation (glutamine, lysine, malate, myristate, ornithine, palmitate, succinate, and aspartate) with the BMI. Finally, the abundance of linoleate, lactate, proline, and isoleucine showed no correlation with BMI (Table 6).

When testing the association between metabolite levels of T2D-sweat EVs and BMI, our data demonstrated a significant direct correlation between BMI and the concentration of EVs isolated from sweat (Table 7). In addition, a positive correlation between all metabolite levels from T2D-sweat EVs with BMI was noted, and for the metabolite succinate, this was statistically significant (Table 7).

Collectively, our data highlight that in healthy participants, there is no evidence of correlation between the abundances of specific metabolites in sweat EVs and BMI; however, there appears to be a positive correlation in T2D participants. For further validation, large-scale studies are required to test this correlation.

## 3. Discussion

In this report, we describe, for the first time, a successful protocol for the isolation and characterization of sweat EVs, using clinically approved dressing patches collected from healthy and T2D participants after heat exposure, while at rest. The size, concentration, and purity of these EVs were examined using different methodologies, thereby providing evidence of their presence and specificity in human sweat.

Our findings demonstrate that sweat EVs carry metabolites associated with a variety of biological pathways, for example amino acids, glutamate, glutathione, fatty acids, TCA, and glycolysis. Metabolites play crucial roles in various physiological cellular processes, such as providing a source of energy, promoting signaling, sustaining cell growth, maintaining cell structure, catalytic activity and defense [16,17,18,19,20]. The number of publications investigating the metabolomics profiling of in vitro-maintained cells (healthy and/or diseased), and from patients in the clinic, has increased significantly in recent years. Furthermore, certain metabolites established as biomarkers are now widely used as a tool for the early diagnosis of numerous pathologies, such as tumors [21,22]. None of these reports have, however, examined the metabolic profile of EVs extracted from sweat. The findings in our study, thus underscore the potential for identifying metabolites in sweat EVs, which may be used as a non-invasive tool to understand the physiological functions of human sweat.

Previously, when analyzing amino acid levels, arginine was reported as the metabolite with the highest concentration present in the sweat of healthy individuals following heat exposure. Additionally, the concentration of arginine in the sweat was independent of its variations in the blood [23]. One explanation of this phenomena is that arginine is metabolized by the sweat glands to produce nitric oxide [24,25,26], and this may enhance the sweat production. Diabetes is characterized by an impaired thermoregulation mechanism that leads to decrease sweating ability, which is notable under heat exposure [27,28,29,30].

Considering the reported data and our findings indicating that sweat EVs contain arginine and that there is a moderate increase of its levels in T2D- versus healthy-sweat EVs, we speculate that arginine levels in EVs extracted from sweat may represent a potential biomarker for studying the sweating capacity of healthy and diabetic individuals notably during exposure to heatwaves. To test this hypothesis, further validation is needed via conducting large-scale clinical trials.

Type 2 diabetes, obesity, and other metabolic disorders are characterized by the accumulation of phospholipids and fatty acids [31,32,33,34]. Our findings indicate that unsaturated and saturated fatty acids, such as linoleate, palmitate, and myristate, are abundant in sweat EVs, and their levels were elevated in T2D-sweat EVs compared to healthy samples following heat exposure. When this initial data are validated through conducting larger-scale population studies, our methodology may indicate that skin patch enriched-sweat EVs could provide early diagnostic solutions for diabetes and obesity, via scoring changes in the levels of those fatty acids compared to their levels in blood.

A recent report has revealed that microbial metabolites are important biomarkers for the risk of Type 2 diabetes [35]. Considering the ability of extracellular vesicles to cross biological barriers, such as the gut, blood and brain [7], as well as the current data highlighting the presence of different metabolites in sweat EVs, we speculate that the non-invasive methodology described here may offer a window to investigate EVs’ transcytosis from internal organs into the sweat by examining their metabolomic composition. Our findings demonstrate a proof-of-concept for a new approach of monitoring health parameters for the early diagnosis of diseases, based on the use of skin patch-enriched sweat EVs.

Our methodology identified lactate as one of the abundant metabolites in sweat EVs in both healthy and T2D participants. Lactate is considered a valuable metabolite for monitoring sport-related physical activity [36,37], and recently, wearable devices have been developed to monitor lactate [3,38,39,40,41]. Taking into consideration our data and the literature, we suggest that once our preliminary observations are confirmed using a larger number of participants, monitoring lactate in sweat EVs may be considered as a groundbreaking strategy to gain insight into the function of sweat glands and performance during training, and notably for managing diabetes.

The strengths of the current study are the novel methodology for enriching sweat EVs non-invasively, the recognition of constituent metabolites of sweat EVs as potential biomarkers, and the proof-of-concept of showing a possible association between blood glucose levels, or BMI, with metabolites’ abundance in sweat EVs.

One major limitation of the current study is the population size, owing to the limited number of participants from healthy and T2D groups. This limitation, and the high variations of metabolite levels in sweat EVs between individuals within the same group, make the statistical power low, therefore, it is difficult to differentiate as to whether the changes in the metabolites’ levels are triggered by the heat or the metabolic state of diabetes. Conducting studies with larger numbers of participants is therefore needed to not only evaluate the metabolite concentrations in the EVs extracted from sweat in response to different stressors such as glucose intervention, heat, and exercise, but also to confirm the association between the metabolite levels in sweat EVs and blood glucose, as well as BMI, in response to the same stressor. If such associations are established, then the metabolites can be used as potential biomarkers for predicting the blood glucose levels in healthy people and individuals with chronic conditions, such as diabetes. Another limitation in this study is the normalization methods used; while the only method used here is particle number/mL, other comparative methods are needed, using different normalizations of the same set of data, for example, that of sweat from the same individual before heat exposure (pilocarpine protocol), and of metabolite levels from healthy sweat gland organoids, as well as further applying enrichment analysis to find the best methods for allowing significant differences to be observed between two distinct groups, such as healthy and diabetes.

Once the limitations described above are addressed, the data collected from this study may pave the way towards developing personalized solutions for measuring and monitoring wellbeing and vital health parameters.

## 4. Material and Methods

### 4.1. Study Design

The controlled study has been approved by the Northern Ostrobothnia Hospital District Ethics committee (EETMK:199/2016). The study is registered in Clinical Trials (NCT02855905). Two groups of participants, healthy (*n* = 11) and T2D (*n* = 10), were recruited. The inclusion criteria for this study were: BMI 25–35, age 40–70 years, HbA1c 7–10% (53–86 mmol/mol), and non-smoking. The participants with T2D were taking different medications for their diabetes (metformin, DPP4 inhibitor, SGLT2 inhibitor, GLP-1 receptor agonist, and long-acting insulin), as well as for comorbid hypertension (diuretic, calcium channel blocker, angiotensin II receptor blocker, beta-blocker, central agonist, and ACE inhibitor) and cholesterol (statins).

Prior to the laboratory study, cardiac autonomic function was evaluated with standard tests. Peripheral neuropathies were assessed by measuring perception thresholds of temperature and pain. Peripheral vascular disease was assessed by blood pressure measurements and by calculating the ankle brachial index (ABI). Touch sensation was assessed by a Semmes–Weinstein monofilament and pressure perception was measured by the use of a tuning fork. Information about kidney function and retinal scans was obtained through regular clinical monitoring.

### 4.2. Heat Exposure Trial in the Laboratory

The controlled experiment was performed at the thermal laboratories of Kastelli Research Center, Oulu, Finland, where 11 healthy controls and 10 T2D participants were exposed to heat (+40°C; RH 40%) under resting conditions, for 90 min. All the baseline and the recovery measurements were obtained at a neutral temperature (+25 °C). Study participants wore light clothing (shorts, t-shirt, and socks). Blood glucose was obtained through blood samples before and after the heat exposure. Total sweat rate was obtained via weighing the participants before and after the experiment. The local sweat rate and quantity was evaluated via weighing the Sorbact^®^ superabsorbent dressing (ABIGO Medical AB—an Essity Company, Askersund, Sweden) collected from the indicated site.

### 4.3. Isolation of Sweat EVs

After collection, the Sorbact^®^ dressing was opened, and the superabsorbent layer was dissolved in 40 mL (1×) PBS, before being filtered through a 40 mm sieve, to remove large particles and pieces of fiber. The filtrate was then re-filtered through 0.8 µm vacuum filter units (Thermo Fisher Scientific, Waltham, MA, USA). The filtrate was centrifuged in a Sorvall AH-629 rotor at 100,000× *g* for 3 h, in a Sorvall Ultracentrifuge Machine WX ultra 90 (VWR, Thermo Electron Corporation, Radnor, PA, USA). The supernatant was collected, the pellets were washed using 1× PBS followed by centrifugation at 100,000× *g* for 2 h, and then were resuspended in 40–60 µL of 1× PBS.

All materials (chemicals, antibodies, filters), and instruments used in the protocol are detailed in the Appendix A.

### 4.4. Targeted LC-MS Metabolomics Analysis

Metabolites were extracted from 10 µL of collected sweat or sweat nanovesicles using 400 µL of cold LC-MS grade extraction solution (Acetonitrile: Methanol:MQ; 40:40:20). Subsequently, samples were vortexed for 2 min and sonicated for 1 min, followed by centrifugation at 14,000× *g* rpm at 4 °C for 5 min. Supernatants were transferred into polypropylene tubes and placed into a nitrogen gas evaporator. The dried samples were suspended in 40 µL of extraction solvent (ACN:MeOH:MQ; 40:40:20) and vortexed for 2 min before being transferred into HPLC glass auto sampler vials. Sample volumes of 2 µL were injected into a Thermo Vanquish UHPLC, coupled with a Q-Exactive Orbitrap quadrupole mass spectrometer that was equipped with a heated electrospray ionization (H-ESI) source probe (Thermo Fisher Scientific, Waltham, MA, USA). A SeQuant ZIC-pHILIC (2.1 × 100 mm, 5 μm particle) column (Merck, Darmstadt, Germany) was used for chromatographic separation. The gradient elution was carried out with a flow rate of 0.100 mL/min using 20 mM ammonium hydrogen carbonate, adjusted to pH 9.4, with ammonium solution (25%) as mobile phase A and acetonitrile as mobile phase B. The gradient elution was initiated from 20% mobile phase A and 80% of mobile phase B and maintained for 2 min, followed by 20% mobile phase A gradually increasing up to 80% until 17 min, then decreasing from 80% to 20% in 17.1 min, which was then maintained up to 24 min. The column oven and auto-sampler temperatures were set to 40 ± 3 °C and 5 ± 3 °C, respectively.

MS was equipped with a heated electrospray ionization (HESI) source using polarity switching and the following settings: a resolution of 35,000, spray voltages of 4250 V for positive and 3250 V for negative mode, the sheath gas, comprising 25 arbitrary units (AU), the auxiliary gas, comprising 15 AU, sweep gas flow 0, a capillary temperature of 275°C, and an S-lens RF level of 50.0. Instrument control was operated with the Xcalibur 4.1.31.9 software (Thermo Fisher Scientific, Waltham, MA, USA). The peak integration was calculated using the TraceFinder 4.1 software (Thermo Fisher Scientific, Waltham, MA, USA) using confirmed retention times for 41 metabolites, standardized with the library kit MSMLS-1EA (Merck). The data quality was monitored throughout the run using serum QC samples interspersed throughout the run, every 10th sample. The metabolite data were checked for peak quality (poor chromatograph), % RSD (20% cutoff) and carryover (20% cutoff).

### 4.5. Metabolomics Data Analysis

The data were analyzed with Metaboanalyst 5.0 (Metaboanalyst, https://www.metaboanalyst.ca, Accessed 17 April 2023). Nanovesicle metabolite peak values were normalized both to total peak intensity and to NV concentration. For multivariate clustering heatmap analysis, the data were statistically interquartile range-filtered and heatmap-computed with distance measures, using Euclidean and clustering algorithms using ward D, with clustering (Figure 3) and without clustering (Figure 5).

### 4.6. Western Blot

The quantity of protein in the EV samples was calculated using the BCA assay (Pierce^TM^ BCA Protein Assay Kit) according to the manufacturer’s recommendations. Equal amounts of total protein (10 µg) were separated by SDS-PAGE and were electrophoretically transferred to nitrocellulose membranes (741280, Macherey-Nagel GmbH & Co., Düren, Germany). The membranes were then incubated overnight at 4 °C with primary antibodies (anti-HSP70 (sc-373867), and anti-CD63 (sc-5275), Santa Cruz Biotechnology, Dallas, TX, USA), acquired from Santa Cruz Biotechnology, and then with secondary antibody (P0260, Aligent Tech, Glostrup, Denmark). The blots were then developed using Pierce^TM^ ECL plus Western blotting substrate (32132, Thermo Fisher Scientific, Waltham, MA, USA).

### 4.7. NanoSight Nanoparticle Tracking Analysis

Nanoparticle tracking analysis (NTA) was performed using a NanoSight NS300 (NanoSight Ltd., Amesbury, UK) equipped with a 405 nm laser. At least four 60 s videos were recorded for each sample, with a camera level of 14 and the detection threshold set at 5. Temperature was monitored throughout the measurements. Videos recorded for each sample were analyzed with NTA software version 3 to determine the concentration and size of measured particles, with the corresponding standard error. For analysis, automatic settings were used for blur, minimum track length and minimum expected particle size. Double-distilled H_2_O was used to dilute the sample.

### 4.8. ExoView Analysis

ExoView^®^ chips coated with anti-CD9 (clone H19a), cd81 (clone JS81), and CD63 (clone H5C6) antibodies (EV-TETRA-C, NanoView Bioscience, Boston, MA, USA) were pre-scanned following the manufacturer’s instructions, in order to measure the baseline signal before the sample incubation. The sweat EVs were diluted in an incubation solution to the quantifiable concentration, to avoid oversaturation of the chip, and 50 µL of sample was carefully loaded into the pre-scanned chip and incubated for 16 h in a sealed 24-well plate (3524, Corning, NY, USA) at RT, on a bench free of vibrations. The chips were washed three times with Solution A for 3 min at 400 rpm at RT on a multiple plate shaker (MPS-1, Biosan, Riga, Latvia), followed by incubation with ExoView Tetraspanin mix (EV-TETRA-C) containing labelled antibodies: anti-CD9 (CF^®^488A), -CD81(CF^®^555), and -CD63 (CF^®^647) in blocking solution for 1 h at RT on an orbital shaker at 400 rpm. Afterwards, the chips were washed once in solution A and three times in solution B, followed by a wash in deionized water and dried following the manufacturer’s instructions. Chips were then scanned with the ExoView R100 reader using the ExoView scanner 2.5.5 acquisition software. Images acquired were analyzed using ExoView Analyzer 3.1.4 software.

### 4.9. Electron Microscopy_Negative Staining

Isolated EVs were settled on a Formvar carbonated copper grid (glow-discharged). Following fixation with 1% glutaraldehyde, the grid was washed with distilled water and stained with neutral 2% UA (Uranyl acetate). Afterwards, the grid was coated with 2% methylcellulose–UA solution, and, following 10 min incubation, the excess fluid was carefully removed and grids were air-dried. EVs were visualized using a Tecnai Spirit G2 transmission electron microscope, and images were taken with a Veleta CCD camera and Item software (Olympus Soft Imaging Solutions GMBH, Munster, Germany).

### 4.10. Immunoelectron Microscopy

Immunoelectron microscopy was performed as previously described [8,42]. EVs samples were settled on a Formvar carbonated grid (glow-discharged). Following incubation in the blocking serum (1% BSA (bovine serum albumin) in PBS), the grids were incubated initially with anti-CD63 antibody (sc-5275, Santa Cruz Biotechnology, Dallas, TX, USA), then with the secondary antibody (AffiniPure Rabbit Anti-Mouse IgG, 315-005-045, Jackson ImmunoResearch Europe Ltd, Cambridgeshire, UK) for 20 min each, and finally with protein A–gold complex (PAG 10 nm) for 20 min. As for the negative controls, four to five samples were incubated using only the secondary antibody. Finally, all grids were stained using neutral uranyl acetate and embedded in methylcellulose/uranyl acetate. All samples were examined using a Tecnai Spirit G2 transmission electron microscope. Images were then captured with a Veleta CCD camera and Item software (Olympus Soft Imaging Solutions GMBH, Munster, Germany).

### 4.11. Statistical Analysis

Spearman’s rank correlation was applied to assess the association of clinical factors with BMI and blood glucose. All analyses were performed with R software version 4.2.0. GraphPad Prism software, version 7, was used for the statistical analyses. The two-tailed Student t-test was employed and * *p*-values less than 0.05 were considered significant.

## 5. Conclusions

Collectively, our study describes a successful methodology for collecting, isolating, and characterizing EVs from sweat, which had been enriched using clinical grade patches. Of particular significance is the demonstration that sweat EVs contain metabolites, the levels of which are different between healthy and diabetic individuals following heat exposure, suggesting that metabolites may represent a way of scoring metabolic changes. The association of metabolite levels with blood glucose and BMI in healthy and Type 2 diabetes groups was distinct; while the abundance of metabolites is significantly associated with blood glucose levels in healthy participants, in Type 2 diabetes, however, the metabolite levels appear to be correlated with BMI. Owing to the limited number of participants and the high variability among individuals, the authors propose to extend the methodology to larger cohorts to further validate these conclusions.

Overall, this study is the first proof-of-concept for using skin patches as a non-invasive way to investigate the physiological behavior of the human body, and this could be applied to different types of clinical studies.

## Figures and Tables

**Figure 1 ijms-24-07507-f001:**
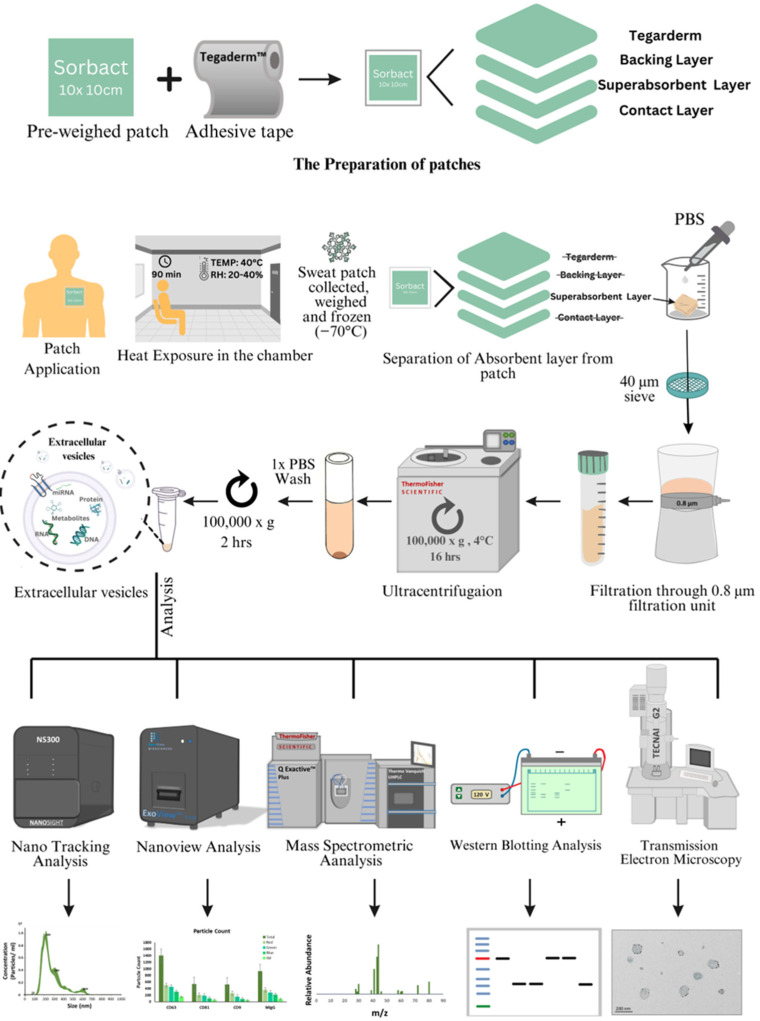
Experimental design. Detailed description of the study design and the sweat EVs’ isolation and characterization steps. RH: Relative Humidity. PBS: phosphate-buffered saline. min: minutes. hrs: hours.

**Figure 2 ijms-24-07507-f002:**
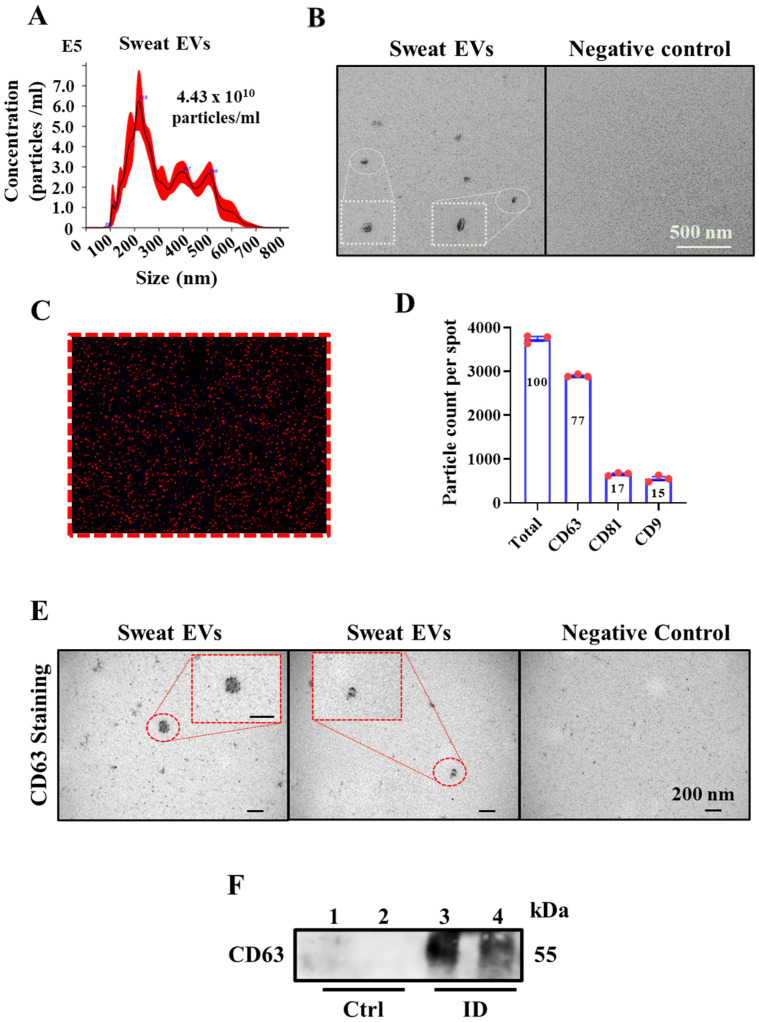
Characterization of sweat EVs. (**A**) NTA analysis of sweat EVs isolated from healthy control participants. (**B**) Electron microscopy analysis. Negative staining of sweat EVs isolated from patches; control refers to the negative patch (was not attached to the patient, but processed in the same way as the positive patches). Scale bar = 200 nm. (**C**,**D**) ExoView analysis of sweat EVs. ExoView image of sweat EVs, stained with antibodies against CD63 (red), CD81 (green), and CD8 (blue), was captured using ExoView R100 reader. (**C**) Particle counts of sweat EVs indicating the percentage of sweat EVs that express CD63, CD81, and CD9. (**D**,**E**) Electron microscopy analysis. Immunostaining of sweat EVs using anti-CD63 antibody; control refers to the negative patch (was attached to the patient, but processed in the same way as the positive patches). Scale bar = 200 nm. (**F**) Western blot analysis of CD63 in sweat EVs isolated from two healthy participants (Lanes 3–4). Control lanes (Ctrl) refer to negative controls, Lane 1: non-enriched sweat patch, that has not been attached to a participant and Lane 2: 1× PBS.

**Figure 3 ijms-24-07507-f003:**
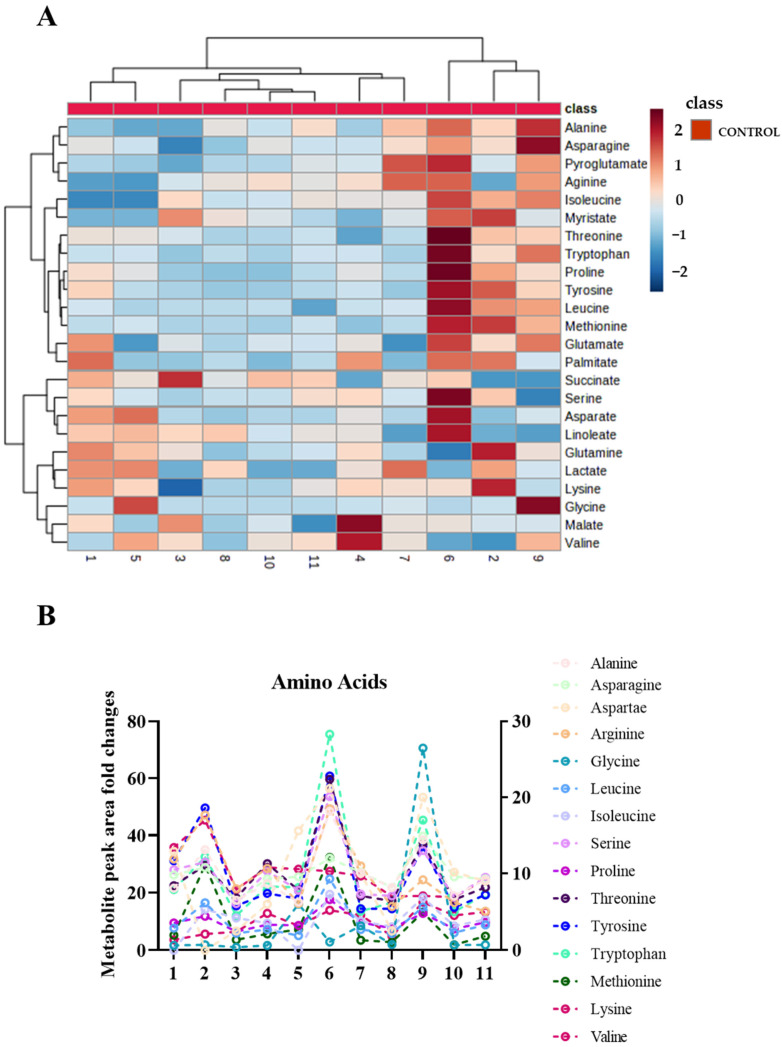
Sweat EVs exhibit metabolic qualities. (**A**) Heatmap illustrating the metabolite pattern of healthy sweat EVs upon heat exposure. (**B**) Metabolite signatures of the amino acid pathway, including alanine, aspartate, arginine, serine, threonine, tyrosine, tryptophan, and lysine, corresponding to the right y-axis, and glycine, leucine, isoleucine, proline, methionine, and valine, corresponding to the left y-axis. Data are metabolite peak area fold changes ± SEM after normalization to the total concentrations of EVs (particles/mL) and to the negative controls (patch without sweat sample). Healthy participants *n* = 11.

**Figure 4 ijms-24-07507-f004:**
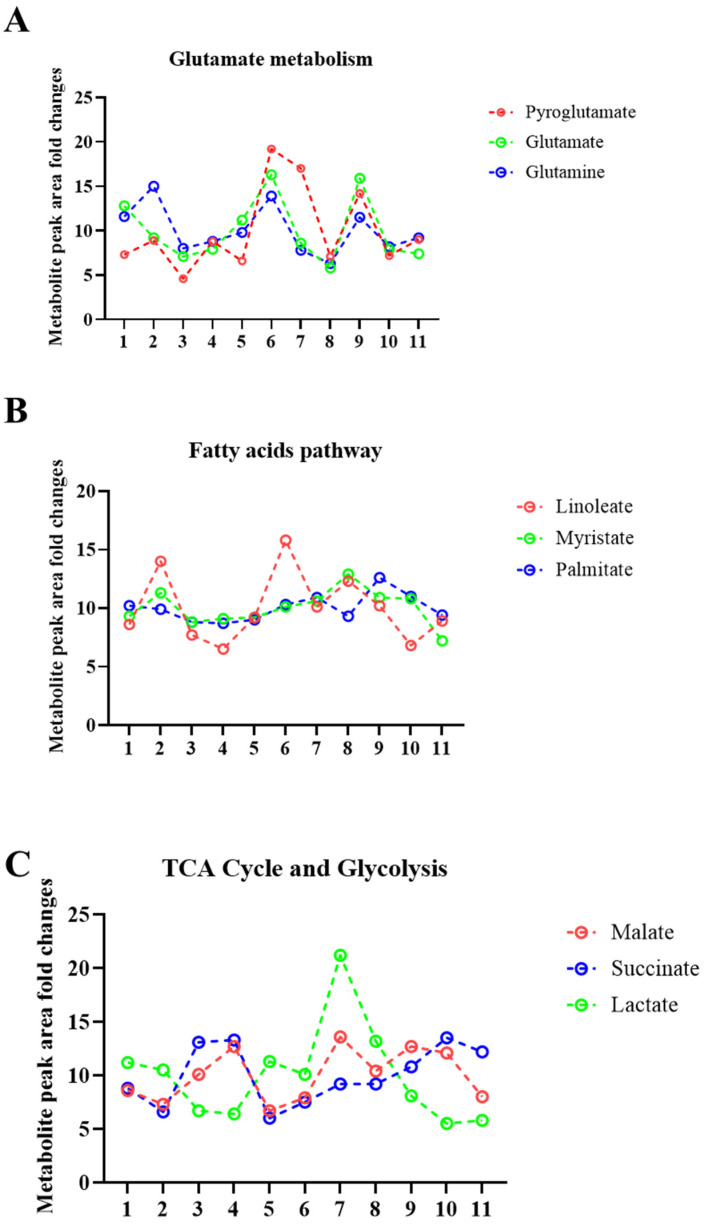
Sweat EVs exhibit metabolic qualities. (**A**) Metabolite signatures of glutathione and glutamate pathways. The metabolites are pyroglutamate, glutamate, and glutamine. (**B**) Metabolite signatures of the fatty acid pathway. The metabolites are linoleate, palmitate, and myristate. (**C**) Metabolite signatures of the TCA cycle and glycolysis pathways. The metabolites are malate, succinate, and lactate. Data are metabolite peak area fold changes ± SEM after normalization to the total EVs concentrations (particles/mL) and to the negative control (patch without sweat sample). *n* = 11.

**Figure 5 ijms-24-07507-f005:**
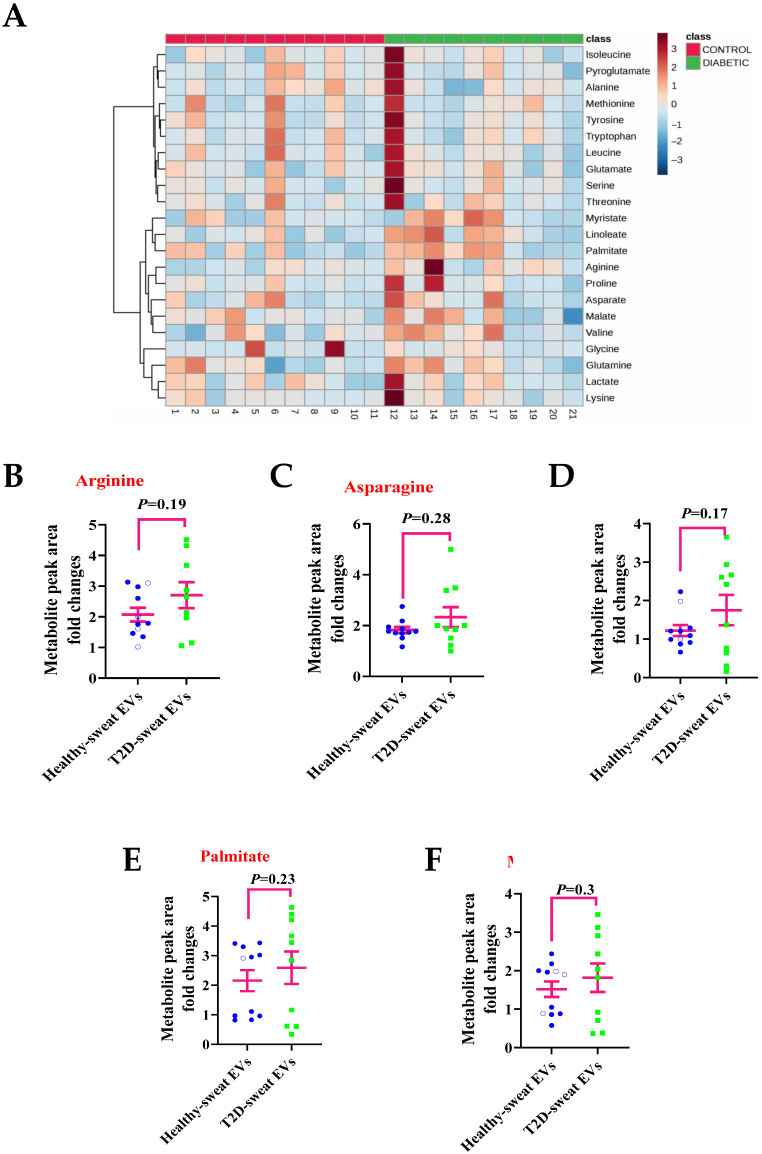
Metabolites from sweat EVs may reflect metabolic changes in diabetes when compared to healthy individuals. (**A**) Heatmap illustrating the metabolite pattern of healthy- and T2D-sweat EVs upon heat exposure. (**B**–**F**) Metabolite levels of Arginine (**B**), Asparagine (**C**), Linoleate (**D**), Palmitate (**E**), and Myristate (**F**). Data are metabolite peak area fold changes ± SEM after normalization to the total EVs concentrations and to the negative control. Two-tailed Student’s *t*-test was used in (**B**) (*p* = 0.19), in (**C**) (*p* = 0.28), in (**D**) (*p* = 0.17), in (**E**) (*p* = 0.23), and in (**F**) (*p* = 0.3). T2D: *n* = 10. Healthy group. *n* = 11.

**Table 1 ijms-24-07507-t001:** Characteristics of the participants in this study. Data indicate means ± SD. Healthy control participants (*n* = 11). N. Number.

Variable	Healthy Participants
Number	11
Gender	Male
Age/years	62.64
Height/cm	176.45 ± 6.74
Weight/Kg	78.87 ± 7.79
Blood glucose/mmol/L	5.7 ± 0.57
Body mass index BMI/Kg/m^2^	25.43 ± 2.04

**Table 2 ijms-24-07507-t002:** Concentration of sweat EVs (particles/mL), blood glucose (BG) levels, and body mass index (BMI) of each participant. *n* = 11 healthy control participants.

	Sweat EV’s Concentration (particles/mL)	Blood Glucose Levels (mmol/L)	Body Mass Index (BMI) (Kg·m^2^)
**Number**			
**1**	4.02 × 10^10^	5.9	25
**2**	2.14 × 10^11^	5	28.5
**3**	1.18 × 10^10^	6	26.3
**4**	2.24 × 10^10^	5.4	23.6
**5**	2.28 × 10^10^	5.4	23
**6**	3.2 × 10^10^	5.4	26
**7**	5.1 × 10^10^	4.5	25
**8**	4.77 × 10^10^	6.2	28
**9**	4.4 × 10^10^	5.2	27.3
**10**	1.42 × 10^11^	4.9	22.3
**11**	6.84 × 10^10^	5.3	24.3

**Table 3 ijms-24-07507-t003:** Characteristics of the participants in this study. Data are means ± SD. T2D participants (*n* = 10).

Variable	T2D Participants
Number	10
Gender	Male
Age/years	64
Height/cm	176.2 ± 5.73
Weight/Kg	93.63 ± 16.28
Blood glucose/mmol/L	8.04 ± 2.58
HbA1c/mmol/mol	49 ± 9.68
Body mass index BMI/Kg/m^2^	30.13 ± 4.83

**Table 4 ijms-24-07507-t004:** Spearman’s rank correlation between blood glucose and metabolite levels in the EVs isolated from healthy control participants. * *p* < 0.05 and ** *p* < 0.005 are significant. ns. not significant.

Variable	Correlation with BG (Healthy)	Lower Band H	Upper Band H	*p* Value
Particles/mL	−0.688	−0.924	−0.073	0.019 *
Pyroglutamate	−0.789	−0.953	−0.269	0.004 **
Glycine	−0.541	−0.873	0.136	0.085 ^ns^
Alanine	−0.807	−0.958	−0.31	0.003**
Arginine	−0.798	−0.955	−0.289	0.003 **
Asparagine	−0.807	−0.958	−0.31	0.003 **
Leucine	−0.615	−0.9	0.039	0.044 *
Glutamate	−0.615	−0.9	0.039	0.044 *
Glutamine	−0.56	−0.881	0.112	0.073 ^ns^
Linoleate	−0.587	−0.89	0.077	0.058 ^ns^
Lactate	−0.495	−0.856	0.189	0.121
Lysine	−0.615	−0.9	0.039	0.044 *
Methionine	−0.642	−0.909	−0.001	0.033 *
Proline	−0.716	−0.932	−0.122	0.013 *
Serine	−0.688	−0.924	−0.073	0.019 *
Threonine	−0.752	−0.943	−0.19	0.008 **
Tyrosine	−0.505	−0.86	0.178	0.113
Malate	−0.716	−0.932	−0.122	0.013 *
Myristate	−0.358	−0.797	0.328	0.28
Valine	−0.697	−0.927	−0.089	0.017 *
Palmitate	−0.321	−0.779	0.361	0.336
Succinate	−0.642	−0.909	−0.001	0.033 *
Tryptophan	−0.606	−0.897	0.051	0.048 *
Aspartate	−0.385	−0.809	0.302	0.242
Isoleucine	−0.817	−0.96	−0.334	0.002 **

**Table 5 ijms-24-07507-t005:** Spearman’s rank correlation between blood glucose (before and after heat exposure) and the metabolite levels in the EVs isolated from T2D volunteers (*n* = 10) upon heat exposure. ns. not significant.

Variable	Correlation with BG (T2D)	Lower Band H	Upper Band H	*p* Value
Particles/mL	0.317	−0.391	0.789	0.372
Pyroglutamate	0.287	−0.418	0.776	0.422
Glycine	−0.165	−0.72	0.519	0.649
Alanine	0.287	−0.418	0.776	0.422
Arginine	0.043	−0.603	0.655	0.907
Asparagine	0.079	−0.579	0.675	0.828
Leucine	0.348	−0.361	0.802	0.325
Glutamate	0.244	−0.456	0.757	0.497
Glutamine	0.079	−0.579	0.675	0.828
Linoleate	0.049	−0.599	0.658	0.894
Lactate	−0.012	−0.637	0.622	0.973
Lysine	0.165	−0.519	0.72	0.649
Methionine	0.628	−0.003	0.901	0.052 ^ns^
Proline	−0.091	−0.682	0.571	0.802
Serine	0.348	−0.361	0.802	0.325
Threonine	0.287	−0.418	0.776	0.422
Tyrosine	0.323	−0.385	0.792	0.362
Malate	−0.012	−0.637	0.622	0.973
Myristate	−0.134	−0.704	0.541	0.712
Valine	−0.116	−0.695	0.554	0.75
Palmitate	−0.433	−0.835	0.27	0.211
Succinate	0.061	−0.591	0.665	0.867
Tryptophan	0.366	−0.343	0.809	0.298
Aspartate	−0.03	−0.647	0.611	0.933
Isoleucine	0.238	−0.461	−0.755	0.508

**Table 6 ijms-24-07507-t006:** Spearman’s rank correlation between BMI and the metabolite levels in the EVs isolated from healthy control participants.

Variable	Correlation with BMI (Healthy)	Lower Band H	Upper Band H	*p* Value
Particles/mL	0.127	−0.514	0.677	0.714
Pyroglutamate	0.118	−0.52	0.672	0.734
Glycine	0.036	−0.576	0.623	0.924
Alanine	0.109	−0.527	0.667	0.755
Arginine	0.164	−0.487	0.698	0.634
Asparagine	0.055	−0.564	0.634	0.881
Leucine	0.182	−0.474	0.708	0.595
Glutamate	0.064	−0.558	0.64	0.86
Glutamine	−0.127	−0.677	0.514	0.714
Linoleate	0.009	−0.594	0.606	0.989
Lactate	0	−0.6	0.6	1
Lysine	−0.155	−0.693	0.494	0.654
Methionine	0.245	−0.425	0.741	0.468
Proline	−0.009	−0.606	0.594	0.989
Serine	0.091	−0.539	0.656	0.797
Threonine	0.018	−0.588	0.611	0.968
Tyrosine	0.145	−0.501	0.687	0.673
Malate	−0.109	−0.667	0.527	0.755
Myristate	−0.164	−0.698	0.487	0.634
Valine	−0.282	−0.67	0.394	0.402
Palmitate	−0.164	−0.698	0.487	0.634
Succinate	−0.136	−0.682	0.507	0.694
Tryptophan	0.291	−0.387	0.765	0.386
Aspartate	−0.027	−0.617	0.582	0.946
Isoleucine	0	−0.6	0.6	1

**Table 7 ijms-24-07507-t007:** Spearman’s rank correlation between BMI and the metabolite levels in the EVs isolated from T2D participants upon heat exposure. * *p* < 0.05 is significant. ns. not significant.

Variable	Correlation with BMI (T2D)	Lower Band H	Upper Band H	*p* Value
Particles/mL	0.77	0.175	0.953	0.014 *
Pyroglutamate	0.527	−0.202	0.88	0.123
Glycine	0.552	−0.172	0.889	0.104
Alanine	0.527	−0.202	0.88	0.123
Arginine	0.345	−0.382	0.808	0.331
Asparagine	0.309	−0.413	0.792	0.387
Leucine	0.442	−0.293	0.849	0.204
Glutamate	0.564	−0.158	0.893	0.096
Glutamine	0.37	−0.36	0.819	0.296
Linoleate	0.127	−0.548	0.702	0.733
Lactate	0.382	−0.35	0.824	0.279
Lysine	0.515	−0.215	0.876	0.133
Methionine	0.309	−0.413	0.792	0.387
Proline	0.394	−0.339	0.829	0.263
Serine	0.442	−0.293	0.849	0.204
Threonine	0.345	−0.382	0.808	0.331
Tyrosine	0.43	−0.304	0.844	0.218
Malate	0.636	−0.061	0.916	0.054 ^ns^
Myristate	0.309	−0.413	0.792	0.387
Palmitate	0.297	−0.423	0.787	0.407
Succinate	0.648	−0.043	0.92	0.049 *
Tryptophan	0.382	−0.35	0.824	0.279
Aspartate	0.152	−0.531	0.715	0.682
Isoleucine	0.333	−0.393	0.803	0.349

## Data Availability

The data sets generated and/or analyzed during the current study are available from the corresponding author on request.

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
