# Peer review of "Clinical-Grade Patches as a Medium for Enrichment of Sweat-Extracellular Vesicles and Facilitating Their Metabolic Analysis"

_ijms, 2023, doi:10.3390/ijms24087507_

Round 1

Reviewer 1 Report

The submitted manuscript described a novel method with clinical grade to collect sweat for EV analysis. The research was interested to readers because the sweat-associated EV studies were not commonly reported, though the significance of sweat-EV as a disease biomarker should be remarked in the field. This study recruited 11 healthy volunteers and analyzed EV-associated metabolites. Authors found certain association with blood glucose or BMI. Overall, the data quality seemed not excellent to me, which may not well supported the conclusion. I would recommend for acceptance with an improved data presentation.

Major:

1. Sweat EV TEM imgaes in Fig2B was not like a normal appearance of EV. I am not sure if these black solid-like particles were resulted from strange negative staining? This data would be suggested to revisited and improved.

2. Western data in Fig2F showing CD63 did not show a good quality. Did control show something there? I would re-run these samples for a better western result.

Minor:

Page 1 line 44: three types of sweat glands: eccrine, apocine, and apocine...again?

Page 3 line 88: loos..typo

Fig2A: unit on Y-axis was particles/ml, or E9 particles/ml? Double check, please.

Reviewer 2 Report

It is an interesting study about the potential use of metabolites inside extracellular vesicles (EVs) from sweat as a source of biomarkers. The study has methodological merit but the experimental design does not allow to draw the conclusions claimed by the authors.

The pellet of extracellular vesicles (lines 310-311) was not washed and the level of contamination with metabolites present in sweat outside extracellular vesicles cannot be estimated. Metabolites present in the supernatant (lines 310-311 and line 96) must be quantified with the same methodology to know if indeed, the EVs contain a peculiar metabolite composition and to know what information provide the set of metabolites in solution and what is provided by EVS.

It is necessary to demonstrate that metabolites inside EVs can reflect changes in the metabolic state of individual participants or groups of participants during health and disease.  In other words, authors must compare the set of metabolites inside EVs between contrasting conditions like fasting Vs after meals, low carbohydrates Vs high carbohydrates diets, healthy VS diabetic or obese cohorts.

It is necessary to compare the information provided by the set of metabolites inside EVs from sweat, with that obtained from the analysis of urine, saliva or other biofluid collected by non-invasive procedures to get a realistic view of the benefits of the proposed methodology.  

Authors show high variability in the response to heat among individuals. If the patches were used in different cohorts, it is not clear How will it be possible to differentiate the changes triggered by heat (90 min, 40°C) and those due to the different metabolic states.

-Figure legend of Figure 3B. Which amino acids correspond to the left and which to the right y-axis?

-Table 1. The units of some variables are missing

-Line126. It is not clear who are “both individuals”

Round 2

Reviewer 2 Report

This manuscript has methodological merit, but it is more appropriate for another journal.

Although extracellular vesicles from sweat have not been extensively studied, the collection procedure (90 min at 40°C) is not very practical in some parts of the world for specimen collection as a source of biomarkers. It is not clear how this approach would be better than collection of other biofluids like urine for example. 
